# Infection via mosquito bite alters Zika virus tissue tropism and replication kinetics in rhesus macaques

Dawn M. Dudley[1], Christina M. Newman[1], Joseph Lalli[2], Laurel M. Stewart[1], Michelle R. Koenig[1], Andrea M. Weiler[3], Matthew R. Semler[1], Gabrielle L. Barry[3], Katie R. Zarbock[1], Mariel S. Mohns[1], Meghan E. Breitbach[1], Nancy Schultz-Darken[3], Eric Peterson[3], Wendy Newton[3], Emma L. Mohr[4], Saverio Capuano III[3], Jorge E. Osorio[2], Shelby L. O'Connor[1,3], David H. O'Connor[1,3], Thomas C. Friedrich[2,3] & Matthew T. Aliota [iD] [2]

Mouse and nonhuman primate models now serve as useful platforms to study Zika virus (ZIKV) pathogenesis, candidate therapies, and vaccines, but they rely on needle inoculation of virus: the effects of mosquito-borne infection on disease outcome have not been explored in these models. Here we show that infection via mosquito bite delays ZIKV replication to peak viral loads in rhesus macaques. Importantly, in mosquito-infected animals ZIKV tissue distribution was limited to hemolymphatic tissues, female reproductive tract tissues, kidney, and liver, potentially emulating key features of human ZIKV infections, most of which are characterized by mild or asymptomatic disease. Furthermore, deep sequencing analysis reveals that ZIKV populations in mosquito-infected monkeys show greater sequence heterogeneity and lower overall diversity than in needle-inoculated animals. This newly developed system will be valuable for studying ZIKV disease because it more closely mimics human infection by mosquito bite than needle-based inoculations.

[1] Department of Pathology and Laboratory Medicine, University of Wisconsin-Madison School of Medicine and Public Health, 3170 UW Medical Foundation Centennial Building, 1685 Highland Ave., Madison, WI 53705, USA. [2] Department of Pathobiological Sciences, University of Wisconsin-Madison School of Veterinary Medicine, 1656 Linden Dr., Madison, WI 53706, USA. [3] Wisconsin National Primate Research Center, University of Wisconsin-Madison, 1220 Capitol Ct., Madison, WI 53715, USA. [4] Department of Pediatrics, University of Wisconsin-Madison, University of Wisconsin Clinical Science Center, 600 Highland Ave, Madison, WI 53792, USA. Correspondence and requests for materials should be addressed to M.T.A. (email: mtaliota@wisc.edu)

Zika virus (ZIKV; *Flaviviridae*, *Flavivirus*) is primarily transmitted by *Aedes aegypti* mosquitoes, but animal models of ZIKV pathogenesis have relied on needle inoculation[1–7]. Needle inoculation has been performed using a range of doses, delivered subcutaneously at a single site or at multiple sites, as well as intravenously, intravaginally, intrarectally, and intra-amniotically in pregnant animals. Each of these routes and inoculum doses could modulate viral infection kinetics and viral tissue distribution on their own, but none of them entirely recapitulate vector delivery of the virus. Blood feeding by a mosquito ensures delivery of the virus to an anatomically precise target in the dermis of the skin[8–10]. When a mosquito feeds it inserts its proboscis into the skin and then actively probes within the tissue for blood. When blood is found, the mosquito begins feeding either directly from the vessel or from the resulting hemorrhage. Importantly, the majority of the inoculum delivered by a mosquito while probing and feeding is deposited extravascularly[11] and only a small amount (~$10^2$ plaque-forming units (PFU)) is deposited intravascularly[12]. Throughout this process, a mosquito injects saliva into the host. The saliva of hematophages, including mosquitoes, is a cocktail of potent pharmacologically active components that prevents clotting and causes vasodilation, as well as alters the inflammatory and immune response, to help facilitate blood feeding[13–15].

Pathogens such as ZIKV exploit this system to infect new vertebrate hosts. Mosquito saliva enhances the replication and pathogenesis of numerous arthropod-borne viruses[16–21]. Furthermore, mosquito saliva has been shown to alter virus dissemination in mammalian hosts for other arboviruses such as dengue virus and Semliki Forest virus[22, 23]. Therefore, saliva delivered to the host by a mosquito may have a critical impact on the initial infection in the skin and may modulate the local innate and adaptive immune response. Accordingly, needle delivery may fail to fully recapitulate important biological parameters of natural ZIKV infection. In addition, the delivery of isolated, purified pathogens by needle inoculation can introduce significant artifacts into the system: for example, directly inoculated virus stocks contain cell culture components not found in mosquito saliva. In sum, it is impossible to replicate the biological, physiological, and mechanical phenomena of mosquito feeding and/or probing using a needle.

The amount of ZIKV inoculated by mosquitoes into a host is not known. In the majority of our previous studies we used an inoculum dose of $1 \times 10^4$ PFU injected subcutaneously (sc)[1, 2, 24], which we chose as a dose likely to be delivered by a ZIKV-infected mosquito. This was based on previous studies, which found that mosquitoes delivered ~$1 \times 10^4$–$1 \times 10^6$ PFU of West Nile virus (WNV)[12] and as much as $1 \times 10^4$ 50% mosquito infectious doses of dengue virus (DENV)[25]. Other recent studies of ZIKV infection in nonhuman primates have relied on a variety of doses and routes with varying outcomes. For example, five sc inoculations each containing $1 \times 10^7$ PFU (a 50-fold higher cumulative dose than any other published study) of a Cambodian strain of ZIKV in a pregnant pigtail macaque (*Macaca nemestrina*) resulted in severe fetal neurodevelopmental abnormalities not seen in other studies using a smaller dose of different ZIKV strains[5]. In rhesus and cynomolgus macaques (*Macaca mulatta* and *Macaca fascicularis*, respectively), ZIKV RNA persisted in saliva and seminal fluids for at least three weeks after clearance of the virus from the peripheral blood following sc inoculation with $1 \times 10^6$ PFU of a Thai ZIKV isolate[26]. Our studies using the same route and doses ranging from $1 \times 10^4$ to $1 \times 10^6$ PFU of a French Polynesian isolate showed no persistence in body fluids after the resolution of acute plasma viremia in non-pregnant macaques[2]. In yet another study in rhesus macaques, intravenous administration of $1 \times 10^5$ PFU of a Brazilian ZIKV isolate resulted in a

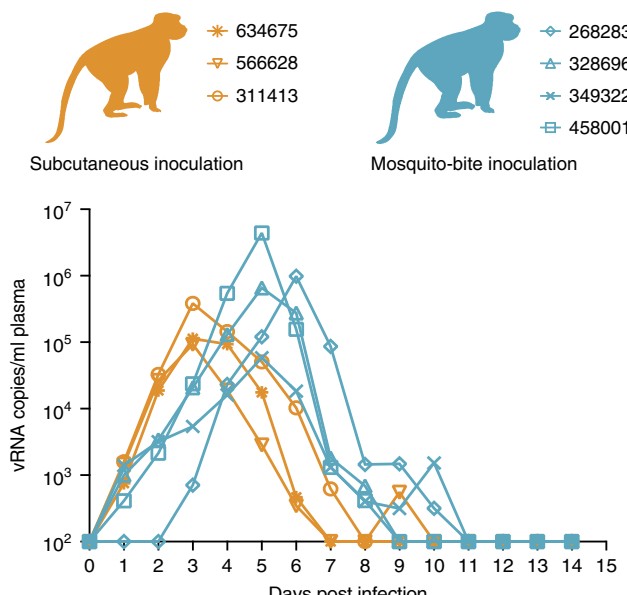

**Fig. 1** Longitudinal detection of Zika vRNA in plasma in subcutaneously inoculated animals (orange) or animals challenged via mosquito bite (blue). Zika vRNA copies per ml blood plasma. The y axis crosses the x axis at the limit of quantification of the qRT–PCR assay (100 vRNA copies/ml)

short-lived plasma viremia and vRNA distribution in a variety of tissues two weeks post-infection[7]. Altogether, these studies established that Asian/American lineage ZIKV infection of rhesus macaques provides a relevant animal model for studying natural history and pathogenesis in a host that has salient similarities to human pregnancy and biology. But they also highlight that differences in route, dose and virus strain may lead to different outcomes. Although no studies to date have addressed whether infection via needle inoculation fundamentally differs from infection via a mosquito vector, transmission of ZIKV via mosquito was attempted in 1956 in Nigeria, but only seroconversion was observed[27].

Here, we show that ZIKV-infected *Ae. aegypti* can reliably initiate systemic ZIKV infections in rhesus macaques. To assess differences in ZIKV replication between virus delivery by needle vs. mosquito vector, we infected rhesus macaques with the Puerto Rican ZIKV isolate PRVABC59 by either sc inoculation ($n = 3$) or by exposure to infected mosquitoes ($n = 4$). All three sc-inoculated macaques were productively infected, with viral load dynamics similar to what we have reported previously[1, 2]. All four animals exposed to *Ae. aegypti* were also productively infected, with noticeable differences in peak viral load and the time to peak viral load.

## Results

**Mosquito-bite delivery of ZIKV results in systemic infection.**
To generate ZIKV-infected mosquitoes, adult female *Ae. aegypti* were fed on ZIKV-infected *Ifnar−/−* mice. Twelve days (d) post-feeding (PF), these same mosquitoes then were allowed to feed on ZIKV-naive macaques. All macaques ($n = 4$) exposed to probing and/or feeding of ZIKV-exposed mosquitoes developed systemic infections as measured by the presence of vRNA in blood plasma (Fig. 1). Since not all mosquitoes bite animals under experimental conditions, we used visual engorgement with blood as a sign of biting and injection of ZIKV. All mosquitoes that fed took a full bloodmeal (indicated by a fully engorged, distended abdomen)

| Table 1 Vector competence of *Aedes aegypti* used to challenge macaques with ZIKV | | |
|---|---|---|
| **12 days post feeding on ZIKV-infected mouse** | | |
| **I** | **D** | **T** |
| 36/40 (90%) | 33/40 (83%) | 10/40 (25%) |
| I, % infected; D, % disseminated; T, % transmitting | | |

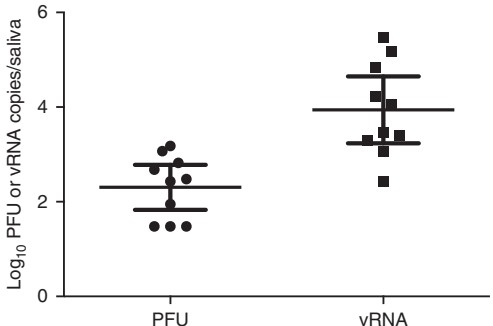

**Fig. 2** Viral titers and RNA loads in saliva of *Aedes aegypti* used to challenge macaques with ZIKV. Mosquitoes were allowed to feed on ZIKV-infected mice. Twelve days later, mosquitoes were exposed to naive macaques. Immediately thereafter, mosquitoes were examined to approximate the amount of virus delivered with mosquito saliva ($n = 40$). Error bars represent 95% confidence interval for the mean

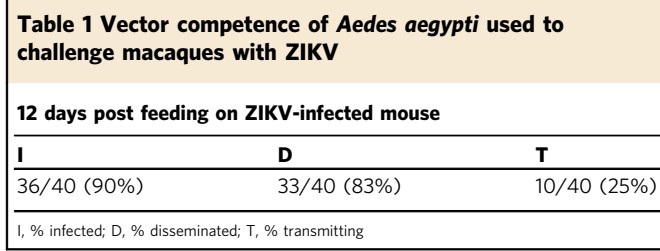

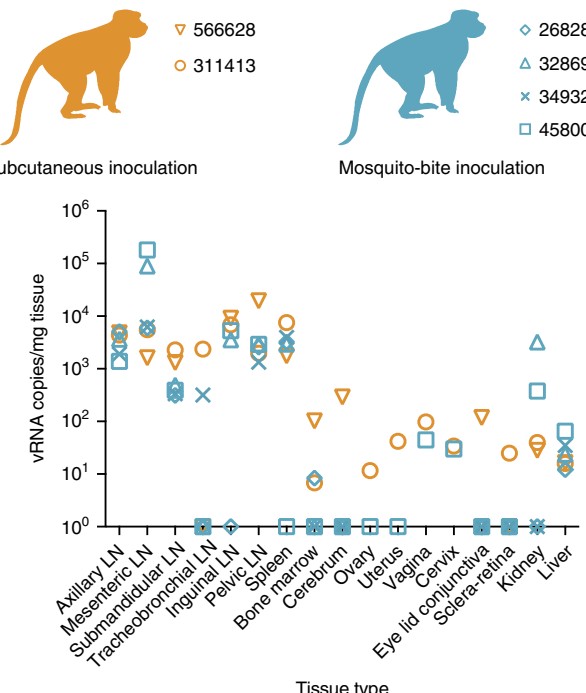

**Fig. 3** Detection of Zika vRNA in tissues in subcutaneously inoculated animals or animals challenged via mosquito bite. qRT–PCR was used to assess the Zika viral burden and tissue distribution in subcutaneously inoculated animals vs. animals challenged via mosquito bite. Orange symbols represent animals infected via subcutaneous injection and blue symbols represent animals that were infected via exposure to ZIKV-infected *Aedes aegypti*. Approximately 24 different tissues were assessed for the presence of viral RNA. Shown are the tissues with positive detection in at least one of the animals per group. The qRT–PCR assay has a quantification threshold of three copies/reaction

and only those mosquitoes that fed also probed. One macaque (328696) received 18 bites, one macaque (349332) received 8 bites, and two macaques (268283 and 458001) received five bites. After macaque feeding, mosquito vector competence for ZIKV was tested on a subset of the same *Ae. aegypti* ($n = 40$) using an in vitro transmission assay[28, 29] at 12 d PF, the same timepoint as the mosquitoes fed on the macaques. As expected, the infection (90%) and dissemination (83%) rates were high for *Ae. aegypti* exposed to ZIKV-infected mice, while the transmission rate was more moderate (25%; Table 1). Infection efficiency indicates the proportion of mosquitoes with virus-positive bodies among the tested ones. Dissemination efficiency indicates the proportion of mosquitoes with virus-positive legs, and transmission efficiency indicates the proportion of mosquitoes with infectious saliva among the tested ones. We used plaque assays on collected saliva to estimate the ZIKV dose inoculated by mosquitoes and found that *Ae. aegypti* saliva titers ranged from $10^{1.5}$ to $10^{3.2}$ PFU (Fig. 2). Importantly, collection of mosquito saliva via capillary feeding does not allow mosquitoes to probe and feed naturally and therefore likely underestimates the dose of virus inoculated. In fact, mosquitoes probing and feeding on a living host delivered doses of other flaviviruses that were 10-fold to 1000-fold higher than those measured using in vitro methods[12].

**Mosquito transmission delays time to peak viral load**. To examine whether viral replication kinetics differed between virus delivery from a needle and delivery from a mosquito vector, we compared viral load dynamics in sc-inoculated animals and animals infected via mosquito bites. There were noticeable differences in peak viral load and the time to peak viral load (Fig. 1). Viral loads in macaques sc-inoculated with $1 \times 10^4$ PFU of ZIKV-PR peaked in all three animals at 3 days post infection (d.p.i.), and ranged from $9.32 \times 10^4$ to $3.85 \times 10^5$ vRNA copies/ml, while

viral loads peaked at 5 or 6 d.p.i. and ranged from $5.83 \times 10^4$ to $4.40 \times 10^6$ vRNA copies/ml in animals that received mosquito bites. Peak viral loads did not differ significantly (Student's *t*-test) between sc-inoculated animals and mosquito-bitten animals ($p$-value = 0.305, *t*-value = 1.144, df = 5); however, peak viral load occurred significantly faster in sc-inoculated animals (linear mixed-effects model; $p$-value = 0.006, *t*-value = −7.606, df = 5). Antibody responses were generated rapidly in both sc-inoculated and mosquito-bitten animals, with detectable changes in IgM as early as 10 d.p.i. (Supplementary Table 1).

**Mosquito transmission alters ZIKV tissue tropism**. To assess whether vRNA levels and tissue tropism differed after virus delivery from needle vs. delivery by mosquito vector, we used quantitative reverse transcriptase PCR (qRT–PCR) to measure ZIKV RNA in homogenized macaque tissues. At 15 d.p.i., both sc-inoculated and mosquito-bite challenged animals were euthanized and tissues collected. In mosquito-bitten animals hemolymphatic tissues contained the highest levels of detectable vRNA. Lymph nodes and spleen were the most highly positive, ranging from $3.1 \times 10^2$ to $1.8 \times 10^5$ vRNA copies/mg of tissue at 15 days PF (Fig. 3). Kidney and liver also were positive; and in the female animal, 458001, reproductive tract tissues were positive (Fig. 3). Notably brain, ocular, and male reproductive tract tissues from all animals contained no detectable vRNA (see Table 2 for a list of all tissues collected and screened). In sc-inoculated animals, hemolymphatic tissues also contained the highest levels of detectable vRNA in both animals (one animal was excluded because tissues

**Table 2 Complete list of tissues examined for ZIKV RNA**

|  | 566628 | 311413 | 268283 | 328696 | 349322 | 458001 |
|---|---|---|---|---|---|---|
| Axillary LN | + | + | + | + | + | + |
| Mesenteric LN | + | + | + | + | + | + |
| Submandibular LN | + | + | + | + | + | + |
| Tracheobroncheal LN | − | + | − | − | + | − |
| Inguinal LN | + | + | − | + | ND | + |
| Pelvic LN | + | + | + | + | + | + |
| Spleen | + | + | + | + | + | − |
| Lung | − | − | − | − | − | − |
| Liver | + | + | + | + | + | + |
| Kidney | + | + | − | + | − | + |
| Bone marrow | + | + | + | − | − | − |
| Cerebrum | + | − | − | − | − | − |
| Eyelid conjunctiva | + | − | − | − | − | − |
| Optic nerve | − | − | − | − | − | − |
| Aqueous humor | − | − | − | − | − | − |
| Sclera retina | − | + | − | − | − | − |
| Cervix | ND | + | ND | ND | ND | + |
| Uterus | ND | + | ND | ND | ND | − |
| Ovarian follicle | ND | − | ND | ND | ND | − |
| Ovary | ND | − | ND | ND | ND | − |
| Vagina | ND | + | ND | ND | ND | + |
| Seminal vesicle | − | ND | − | − | − | ND |
| Testicle | − | ND | − | − | − | ND |
| Prostate | − | ND | − | − | − | ND |

+, ZIKV RNA detected (see Fig. 3 for values); −, ZIKV RNA below the limit of detection; LN, lymph node; ND, no data

were not collected with sterile instruments). Similar to mosquito bitten animals, kidney, liver, and female reproductive tract tissues also contained detectable vRNA; likewise, male reproductive tract tissues contained no detectable ZIKV RNA. In contrast to mosquito bitten animals, ZIKV RNA was detected in the cerebrum of 566628 and the eye of both sc-inoculated animals (Fig. 3).

**Mosquito transmission alters ZIKV populations in macaques.** To characterize the genetic diversity of viral populations transmitted by mosquitoes, we used Illumina deep sequencing to identify single-nucleotide polymorphisms (SNPs) in viral populations present in the viral open reading frame in the ZIKV-PRVABC59 stock, in plasma from monkeys infected by mosquito bite, and in saliva of individual mosquitoes (collected by capillary tubes, $n = 8$; two samples were excluded because of low vRNA concentration) that fed on mice infected with the same virus stock and were verified as transmission competent in our in vitro assay (Fig. 2, Supplementary Table 3). For comparison, we characterized viral SNPs in the 3 monkeys infected by sc injection with $1 \times 10^4$ PFU of the same virus stock. ZIKV sequences from these samples were assembled to a ZIKV-PRVABC59 reference sequence (Genbank KU501215). Across all samples, we detected 42 SNPs occurring in one or more samples at a frequency of ≥5% throughout the viral coding genome (Fig. 4). Of these, 9 SNPs occurred in structural genes, 33 in nonstructural genes. Viral populations in mosquito saliva, and in monkeys infected by these mosquitoes, showed a heterogeneous pattern of SNPs, with fewer SNPs shared among samples (Fig. 4a, b). By contrast, in sc-infected monkeys, the frequency and distribution of viral SNPs were highly similar among infected animals and closely resembled those observed in the stock virus (Fig. 4c). Furthermore, the process of mosquito transmission appeared to alter the frequencies of SNPs observed in the ZIKV-PRVABC59 stock virus: for example, a mutation at reference nucleotide 3147 causing a methionine-to-threonine substitution at position 220 of NS1 (NS1 M220T) was at high frequency in multiple mosquito

samples and in all monkeys infected by mosquito bite, but it remained below 15% frequency in the virus stock and in all animals infected by sc injection. Conversely, a C-to-T mutation at nucleotide 5679 encoding NS3 S356F was present in ~65% of stock viruses and in 40–60% of viruses infecting sc-inoculated animals, but this same mutation was detected in only one mosquito saliva sample and was absent from all animals infected by mosquito bite. We cannot determine from our data whether these changes in SNP frequencies are the result of natural selection or other processes, like genetic drift or founder effects.

Mosquitoes feed on small volumes of blood from infected hosts, limiting the size of the viral population founding infection in the vector. Also, during replication in mosquitoes, flaviviruses undergo population bottlenecks as they traverse physical barriers like the midgut[30, 31]. We therefore reasoned that viral genetic diversity in mosquitoes and monkeys infected by mosquito bite would be low as compared with the virus stock and populations in sc-inoculated monkeys. To test this prediction, we measured within-host viral diversity using the π statistic, which quantifies the number of pairwise differences between sequences without regard to a specific reference. We calculated π for each gene encoding a mature viral protein in each sample (except for protein 2 K and NS4A, where coverage was not deep enough in each sample to allow for rigorous comparisons). Consistent with our expectations, our analysis showed that viral diversity tended to be lowest in mosquito saliva, and highest in the virus stock and sc-inoculated animals (Supplementary Fig. 1). Differences in diversity were most pronounced in the capsid gene, where π was significantly lower in mosquito saliva samples and mosquito-infected monkeys than in monkeys inoculated sc, but significant differences between groups were also found in most other viral genes (Supplementary Fig. 1). Finally, we asked how natural selection might be shaping virus populations in infected mosquitoes and monkeys. The magnitude and direction of natural selection on virus populations can be inferred by comparing within-host levels of synonymous and non-

synonymous nucleotide diversity, denoted respectively as πS and πN. In general, πS > πN indicates that purifying selection is acting to remove deleterious mutations, while πS < πN indicates that positive or diversifying selection is acting to favor the outgrowth of new viral variants. We therefore compared the magnitude of πN and πS in each viral gene across experimental groups. πS was significantly greater than πN in multiple nonstructural genes in sc-inoculated monkeys, suggesting that virus populations were

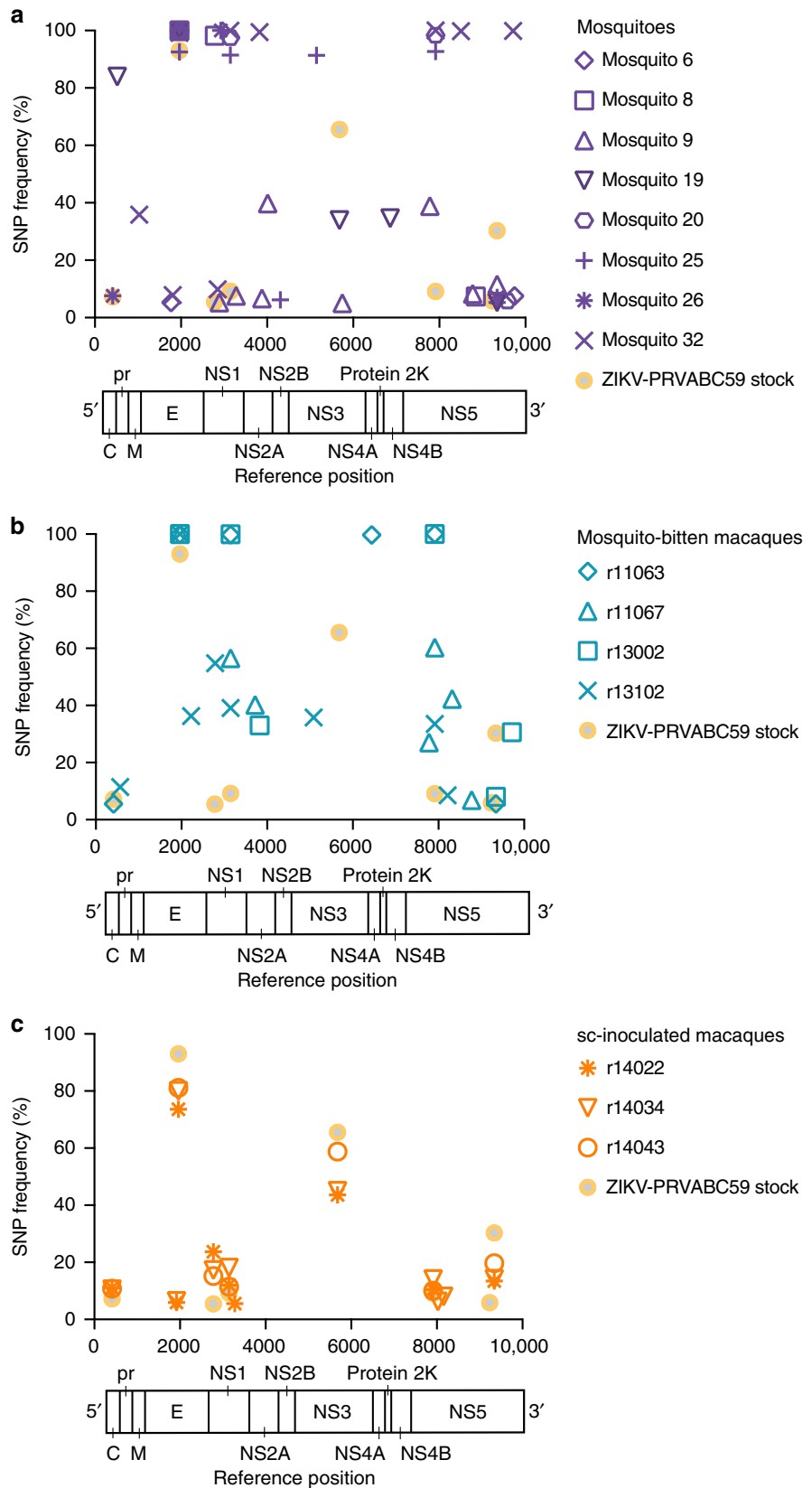

largely under purifying selection (Supplementary Fig. 2). In contrast, we find evidence for purifying selection in mosquito-infected monkeys only in capsid and NS5, and only in NS5 in mosquito saliva. Taken together, our data show that ZIKV populations in mosquito-infected monkeys exhibit more inter-host variability (i.e., different combinations of SNPs) than in sc-inoculated monkeys, perhaps due to sharp viral population reductions that occur as virus passes from one vertebrate host, through physical barriers in the mosquito, and to a new host. We do not find evidence for strong natural selection acting at the gene level in mosquitoes or mosquito-infected monkeys in our data, suggesting that reduced diversity results from random population bottlenecks rather than selection for particular variants.

**Viremic rhesus macaques do not infect mosquitoes**. To better understand ZIKV transmission dynamics, *Ae. aegypti* vector competence for ZIKV from macaques was evaluated at days 7, 13, and 25 d PF from mosquitoes that fed on sc-inoculated animals at 3 d.p.i. and days 13 and 25 from mosquitoes that fed on mosquito-infected animals at 4 d.p.i. All samples from mosquitoes that fed on sc-inoculated animals were negative for ZIKV by plaque assay at all timepoints (Table 3). A single *Ae. aegypti* that fed on animals infected by mosquito bite had a disseminated ZIKV infection at d 25 PF, but this mosquito was not capable of transmitting the virus as measured by plaque assay (Table 3). These data are consistent with field epidemiological reports, which estimated mosquito infection rates during ZIKV outbreaks to be 0.061%[32] and also are consistent with infection rates during DENV and chikungunya outbreaks[33]. It should also be noted that *Ae. aegypti* with poor competence but high population density have been capable of sustaining outbreaks of arboviral diseases such as yellow fever[34].

## Discussion

In 1956 in Nigeria, investigators attempted to transmit ZIKV to a healthy rhesus monkey by exposing it to mosquitoes that had fed on ZIKV-infected mouse blood. The animal remained healthy, and no virus was detected in blood, but the animal apparently developed antibodies to ZIKV (no data were shown). As a result, it was concluded that ZIKV transmission was achieved[27]. But beyond this single manuscript, mosquito-bite delivery of ZIKV for laboratory studies has not been reported in the literature. Here we demonstrate, for the first time, to our knowledge, that systemic infection in nonhuman primates can result from mosquito-bite delivery of ZIKV. This work thus establishes a nonhuman primate model for vector-borne ZIKV transmission. Using this model, we observed a significant delay to peak viremia when virus was delivered by mosquito as compared to sc needle inoculation. While viral genetic composition and diversity levels in sc-inoculated monkeys largely mirrored those of the stock virus, we observed a trend toward decreased genetic diversity in virus populations infecting mosquitoes and the monkeys infected by them. Viral passage from mice through mosquitoes to macaques appeared to alter the frequencies of SNPs in the ZIKV population as compared with the stock virus. We found no evidence that selection favored specific variants in mosquito transmission;

instead, we observed a more heterogeneous distribution of viral SNPs in mosquito-infected monkeys than in sc-inoculated animals. These findings are consistent with the observation that arboviruses undergo multiple population bottlenecks in infected mosquitoes[30, 31]; as a result, different mosquitoes could transmit different founder populations to different macaques. Furthermore, because each macaque was bitten by multiple mosquitoes, viral populations in macaques could represent mixtures of populations from more than one mosquito. The low number of viral genome copies isolated from mosquito saliva, together with the differing patterns of SNPs observed among mosquito-bitten macaques, are also consistent with the interpretation that transmission between mammalian hosts via mosquitoes involves one or more random viral population bottlenecks. A recent study of DENV evolution in mosquitoes found evidence for both random population bottlenecks and purifying selection as the virus replicated in mosquitoes and crossed anatomical barriers[30]. Our data reveal only limited evidence of purifying selection on ZIKV populations sequenced from mosquito saliva (indeed, we observed the strongest signals of purifying selection in viruses infecting sc-inoculated monkeys), but our study investigated virus populations at single timepoints in infected macaques and mosquitoes and was therefore not powered to carefully evaluate within-host selection on virus populations over time. There is no evidence in our data suggesting that differences in the genetic composition of viruses infecting mosquito-bitten macaques resulted in phenotypic differences in, e.g., disease severity or dissemination of virus to different tissues, but if the source virus population encoded broader phenotypic diversity it is possible that founder effects associated with mosquito transmission could result in phenotypically different infections in different hosts. Mosquitoes were infected by feeding on ZIKV-infected immune-deficient mice; it is therefore possible that viral variants emerging during replication in mice could have encoded some of the phenotypic differences we observed in macaques. However, this appears unlikely, as no specific SNP or constellation of SNPs was consistently detected that distinguished viruses replicating in mosquito-bitten animals from viruses present in sc-inoculated animals.

It is well established that mosquito transmission can affect arbovirus infection outcomes, but outcomes vary depending on the mosquito-virus-host system[11, 12, 17, 35]. In this study we found a single sc dose of $1 \times 10^4$ PFU of ZIKV led to an altered tissue distribution as compared to mosquito-inoculated animals at 15 d.p.i. Although a single sc inoculation may not perfectly model vector-borne transmission of ZIKV, we anticipate that sc inoculation will continue to be valuable for studying ZIKV pathogenesis. Needle inoculation remains an important option when it is necessary to control the exact dose delivered, but the dose and number of sc inoculations could alter virus distribution and replication kinetics. For example, several recent studies using needle inoculation detected a broad ZIKV RNA tissue distribution, including in reproductive organs, the CNS, and brain[7, 26, 36, 37]. Likewise, we also found evidence of ZIKV infection of the brain and eye in our needle-inoculated animals. In contrast, animals challenged by mosquito bite had no evidence of infection of the

**Fig. 4** Single-nucleotide polymorphisms in ZIKV populations infecting monkeys and mosquitoes in this study. We used Illumina deep sequencing to characterize viral genomic diversity in **a** saliva samples from mosquitoes that fed on ZIKV-infected *Ifnar−/−* mice, **b** monkeys infected via mosquito bite, and **c** monkeys inoculated subcutaneously with ZIKV-PRVABC59 stock. Purple symbols represent individual saliva samples, blue symbols represent animals that were infected via exposure to ZIKV-infected *Aedes aegypti*, and orange symbols represent animals infected via subcutaneous injection. Frequencies of SNPs detected in the stock virus isolate are plotted in each panel for reference. Viruses infecting monkeys were sequenced at the time of peak plasma viremia: day 3 post-infection in sc-inoculated animals and days 5 or 6 post-feeding in mosquito-bitten animals. Plotted frequencies represent the average of 2 technical replicates for each sample

| Table 3 Vector competence of *Aedes aegypti* following peroral exposure to ZIKV-infected macaques | | | | | | | | |
|---|---|---|---|---|---|---|---|---|
| **Zika virus** | **7 days post feeding** | | | **13 days post feeding** | | | **25 days post feeding** | | |
| | **I** | **D** | **T** | **I** | **D** | **T** | **I** | **D** | **T** |
| Needle-inoculated | 0/30 (0%) | 0/30 (0%) | 0/30 (0%) | 0/40 (0%) | 0/40 (0%) | 0/40 (0%) | 0/29 (0%) | 0/29 (0%) | 0/29 (0%) |
| Mosquito-bite | ND | ND | ND | 0/30 (0%) | 0/30 (0%) | 0/30 (0%) | 1/26 (4%) | 1/26 (4%) | 0/26 (0%) |

I, % infected; D, % disseminated; T, % transmitting; ND, no data
Needle-inoculated animals were exposed to mosquitoes at 3 d.p.i. and mosquito-bite animals were exposed to mosquitoes at 4 d.p.i

brain or CNS at 15 d.p.i. Furthermore, similar to what has been described for human infections (i.e., that most cases are asymptomatic), none of the animals studied here displayed any of the hallmark symptoms (e.g., fever, maculopapular rash, conjunctivitis, etc.) that have been associated with overt Zika fever or those that have been associated with neurologic disease in adults in rare instances (e.g., Guillain-Barré syndrome[38, 39], encephalitis, meningoencephalitis, and myelitis[40]); therefore, outcomes observed here may more closely parallel what has been reported during adult human infections. Finally, we did not detect ZIKV RNA in male reproductive tract tissues. This was somewhat surprising considering reports of both male-to-female and male-to-male sexual transmission of ZIKV[41], both of which are suggestive of the male urogenital tract tissues serving as potential reservoirs for the virus. It is possible that additional analyses of more animals at different timepoints using more careful sampling would more sensitively detect ZIKV RNA.

It is possible that the differences in disease outcomes and virus distribution observed in our study relative to the other studies may be due to the use of a specific strain, dose, or host species. However, many of the published studies used the same Puerto Rican ZIKV strain used in our study. Asian/American lineage outbreak strains share >99% genome-wide nucleotide identity[42], but virus stocks prepared at different centers, while nominally the same strain, have different passage histories, which could result in small, but biologically important, genotypic and phenotypic differences. Another possible explanation for the disparity in outcomes is the inoculation route. For example, iv inoculation likely resulted in antigen presentation to many lymph nodes simultaneously, likely promoting faster innate immune responses and faster clearance of virus from the peripheral blood[7]. Sc inoculation, by contrast, likely resulted in slower dissemination through the draining lymph nodes; dissemination may be slower still following mosquito infection, as indicated by the delay in peak viral loads observed with mosquito-bite delivery of virus in this study. In addition, the use of multiple simultaneous injections[36] could iatrogenically cause disease signs[17]. For example, the fever response observed with multiple injections might be expected, given that injection of virus stocks also delivers cell culture medium components that might serve as irritants; needle puncture of the skin is also more traumatic than insertion of a mosquito proboscis. Accordingly, needle inoculation of DENV resulted in significantly elevated temperatures in the humanized mouse model as compared to mice infected with DENV via mosquito bite[17]. Notably, our sc-inoculated animals did not have elevated body temperatures.

It also is possible that the differences in infection kinetics, tissue distribution, and viral population diversity in mosquito-bitten animals relevant to sc-inoculated animals may be due to the fact that feeding and probing mosquitoes deliver variable inoculum doses. We estimated the infectious ZIKV dose delivered by a mosquito to be $10^{1.5}$ to $10^{3.2}$ PFU per mosquito. In addition, each animal received multiple mosquito bites. Still, with our observed transmission frequency at 25% (Table 1) it is likely that each animal received a cumulative dose lower than $1 \times 10^4$ PFU,

i.e., lower than the dose delivered by needle inoculation. However, as mentioned previously, collection of mosquito saliva via capillary feeding does not allow mosquitoes to probe and feed naturally and therefore likely underestimates the dose of virus inoculated. In fact, mosquitoes probing and feeding on a living host delivered doses of other flaviviruses that were 10- to 1000-fold higher than those measured using in vitro methods[12]. Therefore, it also is possible that the mosquito-bitten macaques received a cumulative dose that was several orders of magnitude higher than that delivered by sc inoculation, but this requires further experimental verification. Importantly, the highest plasma viral loads were observed in the two animals (458001 and 268283) that received the least number of mosquito bites (five bites each). At the very least, these data warrant further exploration into the exact dose of ZIKV delivered by a feeding and probing mosquito.

Surprisingly, in our experiments no *Ae. aegypti* (with the exception of a single mosquito that fed on a mosquito-bite inoculated animal) became infected with ZIKV after feeding on ZIKV-viremic macaques. This was likely the result of the low amount of infectious virus in macaque blood. Our previous studies indicate that the PFU:particle ratio for ZIKV in our system is ~1:1000 and therefore infectious bloodmeal titers for mosquito feeding experiments were <4.0 $\log_{10}$ PFU/ml (ref. [2]). It is likely that mosquito vectors do not become efficiently infected when ZIKV titers are low, and higher viral titers in the bloodmeal increase the probability of mosquito infection[43]. The exact threshold viremia that results in productive mosquito infection remains unknown, but a recent study used artificial membrane feeding to establish a minimum infective dose of 4.2 $\log_{10}$ PFU/ml for susceptibility in mosquitoes[43]. It should be noted that viral loads in macaque plasma resemble those reported in humans in endemic areas[32, 44–47]. Furthermore, the first clinical description of a patient suffering from Zika fever was reported in 1956, and was based on a ZIKV infection experimentally induced in a human volunteer[48]. The patient was a 34-year-old European male infected sc with 265 50% mouse lethal doses of the strain of ZIKV isolated in Nigeria in 1954. His first symptoms were fever and a slight headache 3.5 days after inoculation. The headache lasted approximately two days. A rash was not recorded. The patient also was exposed to female *Ae. aegypti* mosquitoes during the acute stage of illness, and similar to what is described here, ZIKV was not recovered from these mosquitoes, perhaps due to low viremia[48].

We established a mosquito infection model of ZIKV in non-human primates to understand the impact of mosquito transmission on ZIKV pathogenesis. Here we used the rhesus macaque because it is a well-studied translational model for viral pathogenesis and for preclinical evaluation of countermeasures, including during pregnancy. However, the approaches we describe to achieve mosquito transmission of ZIKV in the laboratory have many other applications. For example, mosquito transmission models using New-World NHP could help predict the likelihood of establishing sylvatic ZIKV cycles in the Americas. Parallel studies should also evaluate vector competence of New-World mosquito species, particularly those vectors that may

have the capacity to maintain sylvatic cycles (e.g., *Sabethes* spp. and *Haemogogus* spp.) and those that may be capable of bridging sylvatic and urban cycles (e.g., *Aedes albopictus*)[49]. These mosquito species may have a lower threshold to infection as compared to *Ae. aegypti*, but this will require further laboratory confirmation.

Needle inoculation of arboviruses does not faithfully recapitulate the complex factors involved in vector-borne transmission, which can have important impacts on disease pathogenesis. In our study 4/4 animals were infected in a single feeding session, suggesting that mosquito delivery of ZIKV in nonhuman primates provides a tractable animal model of natural transmission that can be applied to many other mosquito-borne pathogens. For example, studies comparing the course of infection and the immune response between DENVs delivered by mosquito vs. needle could be important for defining the quality of the immune response to dengue where both the mosquito vector and enhancing antibodies should be considered[18]. Ultimately, these same approaches may be useful for testing the safety and efficacy of vaccine and therapeutic candidates: previous reports have demonstrated that *Leishmania* vaccines that protected against needle challenge failed against sandfly bite challenge[50, 51] and a blood-stage malaria vaccine was shown to be ineffective against mosquito-bite challenge in humans[52]. Whether mosquito-vectored ZIKV challenge might alter the efficacy of vaccines relative to needle inoculation remains unknown, but these results from other systems underscore the importance of studying pathogenic outcomes following natural exposure to a pathogen. Eventually the approaches we describe could even be extended beyond mosquitoes to pathogens whose vectors share similar blood feeding strategies, such as tick-borne viruses.

## Methods

**Study design.** This study was designed as a proof of concept study to examine whether sc inoculation of ZIKV fundamentally differs from mosquito bite delivery of ZIKV in the rhesus macaque model. Available animals were allocated into experimental groups randomly with both groups containing male and female animals. Investigators were not blinded to experimental groups.

**Ethical approval.** This study was approved by the University of Wisconsin-Madison Institutional Animal Care and Use Committee (Animal Care and Use Protocol Number G005401 and V5519).

**Nonhuman primates.** Five males and two females, Indian-origin rhesus macaques utilized in this study were cared for by the staff at the Wisconsin National Primate Research Center (WNPRC) in accordance with the regulations, guidelines, and recommendations outlined in the Animal Welfare Act, the Guide for the Care and Use of Laboratory Animals, and the Weatherall report. In addition, all macaques utilized in the study were free of Macacine herpesvirus 1, Simian Retrovirus Type D, Simian T-lymphotropic virus Type 1, and Simian Immunodeficiency Virus. For all procedures, animals were anesthetized with an intramuscular dose of ketamine (10 ml/kg). Blood samples were obtained using a vacutainer or needle and syringe from the femoral or saphenous vein.

**Cells and viruses.** African Green Monkey kidney cells (Vero; ATCC #CCL-81) were maintained in Dulbecco's modified Eagle medium (DMEM) supplemented with 10% fetal bovine serum (FBS; Hyclone, Logan, UT), 2 mM L-glutamine, 1.5 g/l sodium bicarbonate, 100 U/ml penicillin, 100 µg/ml of streptomycin, and incubated at 37 °C in 5% $CO_2$. *Aedes albopictus* mosquito cells were (C6/36; ATCC #CRL-1660) were maintained in DMEM supplemented with 10% fetal bovine serum (FBS; Hyclone, Logan, UT), 2 mM L-glutamine, 1.5 g/l sodium bicarbonate, 100 U/ml penicillin, 100 µg/ml of streptomycin, and incubated at 28 °C in 5% $CO_2$. ZIKV strain PRVABC59 (ZIKV-PR; GenBank:KU501215), originally isolated from a traveler to Puerto Rico with three rounds of amplification on Vero cells, was obtained from Brandy Russell (CDC, Ft. Collins, CO). Virus stocks were prepared by inoculation onto a confluent monolayer of C6/36 mosquito cells with two rounds of amplification. A single harvest with a titer of $1.58 \times 10^7$ plaque-forming units (PFU) per ml (equivalent to $2.01 \times 10^{10}$ vRNA copies per ml) of Zika virus/H. sapiens-tc/PUR/2015/PRVABC59-v3c2 were used for challenges. We deep sequenced the challenge stock to verify the expected origin (see details in a section below). The ZIKV challenge stock sequence matched the GenBank sequence (KU501215) of the parental virus, but there were five sites where between 5 and

92% of sequences contained variants that appear to be authentic (four out of five were non-synonymous changes; Supplementary Table 2).

**Mosquito strains and colony maintenance.** The *Aedes aegypti* black-eyed Liverpool (LVP) strain used in this study was obtained from Lyric Bartholomay (University of Wisconsin-Madison, Madison, WI) and maintained at the University of Wisconsin-Madison as previously described[53]. *Ae. aegypti* LVP are ZIKV transmission competent[28].

**Subcutaneous inoculations.** The ZIKV-PR stock was thawed, diluted in PBS to $1 \times 10^4$ PFU/ml, and loaded into a 3 ml syringe maintained on ice until inoculation. For sc inoculations, each of three Indian-origin rhesus macaques was anesthetized and inoculated sc over the cranial dorsum with 1 ml virus stock containing $1 \times 10^4$ PFU. All animals were closely monitored by veterinary and animal care staff for adverse reactions and signs of disease. Animals were examined, and blood, urine, oral swabs, and saliva were collected from each animal daily from 1 through 10 days, and 14 d.p.i., and beginning on the fifteenth day animals were humanely killed and necropsied.

**Mosquito bite challenges.** Mosquitoes were exposed to ZIKV by feeding on isoflurane anesthetized ZIKV-infected *Ifnar−/−* mice, which develop sufficiently high ZIKV viremia to infect mosquitoes[28, 29]. *Ifnar−/−* on the C57BL/6 background were bred in the pathogen-free animal facilities of the University of Wisconsin-Madison School of Veterinary Medicine. Two male and two female four-week-old mice were used for mosquito exposures. Mice were inoculated in the left, hind foot pad with $1 \times 10^6$ PFU of ZIKV in 50 µl of sterile PBS. Three- to six-day-old female mosquitoes were allowed to feed on mice two days post infection at which time sub-mandibular blood draws were performed and serum was collected to verify viremia. These mice yielded an infectious bloodmeal concentration of 5.64 $\log_{10}$ PFU per ml ± 0.152 (mean ± s.d.). Mosquitoes that fed to repletion were randomized, separated into cartons in groups of 10–50, and maintained on 0.3 M sucrose in an environmental chamber at 26.5 °C ± 1 °C, 75% ± 5% relative humidity, and with a 12 h photoperiod within the Department of Pathobiological Sciences BSL3 Insectary Facility at the University of Wisconsin-Madison. Eleven days later, following oviposition, ZIKV-exposed mosquitoes were sucrose starved for 14 to 16 h and on the twelfth day mosquitoes were exposed to naive, ketamine anesthetized macaques. The mesh top of a 0.6 liter carton containing 10–50 ZIKV-exposed mosquitoes (numbers varied to ensure feeding success) was placed in contact with the left forearm of each of four macaques. The forearm was chosen because there was little hair to obstruct mosquito feeding, it could easily be placed on top of the mosquito carton, and mosquitoes could be easily monitored during the feeding. Mosquitoes were allowed to probe and feed on the forearm for five minutes. Mosquitoes were monitored during feedings and the number of mosquitoes that probed and the blood engorgement status of each mosquito were recorded. ZIKV infection, dissemination, and transmission status was confirmed in a subset of 40 mosquitoes as described in a subsequent section. As described previously, animals were examined, and blood, urine, and oral swabs were collected from each animal daily during challenge. None of the mosquito-bite challenged animals exhibited cutaneous reactions to mosquito bites and bites were not associated with itching.

**Vector competence.** Infection, dissemination, and transmission rates were determined for individual mosquitoes and sample sizes were chosen using long established procedures[28, 29, 44]. Mosquitoes that fed to repletion on macaques were randomized and separated into cartons in groups of 40–50 and maintained as described in a previous section. Mosquitoes were exposed to the sc-inoculated animals at 3 d.p.i. and the mosquito-bitten animals at 4 d.p.i. All samples were screened by plaque assay on Vero cells. Dissemination was indicated by virus-positive legs. Transmission was defined as release of infectious virus with salivary secretions, i.e., the potential to infect another host, and was indicated by virus-positive salivary secretions.

**Plaque assay.** All ZIKV screens from mosquito tissues and titrations for virus quantification from mouse serum or virus stocks were completed by plaque assay on Vero cell cultures. Duplicate wells were infected with 0.1 ml aliquots from serial 10-fold dilutions in growth media and virus was adsorbed for one hour. Following incubation, the inoculum was removed, and monolayers were overlaid with 3 ml containing a 1:1 mixture of 1.2% oxoid agar and 2X DMEM (Gibco, Carlsbad, CA) with 10% (vol/vol) FBS and 2% (vol/vol) penicillin/streptomycin. Cells were incubated at 37 °C in 5% $CO_2$ for four days for plaque development. Cell monolayers then were stained with 3 ml of overlay containing a 1:1 mixture of 1.2% oxoid agar and 2× DMEM with 2% (vol/vol) FBS, 2% (vol/vol) penicillin/streptomycin, and 0.33% neutral red (Gibco). Cells were incubated overnight at 37 °C and plaques were counted.

**Viral RNA isolation.** Plasma was isolated from EDTA-anticoagulated whole blood collected the same day by Ficoll density centrifugation at 1860 rcf for 30 minutes. Plasma was removed to a clean 15 ml conical tube and centrifuged at 670 rcf for an additional 8 minutes to remove residual cells. Viral RNA was extracted from 300 µl

plasma using the Viral Total Nucleic Acid Kit (Promega, Madison, WI) on a Maxwell 16 MDx instrument (Promega). Tissues were processed with RNAlater (Invitrogen, Carlsbad, CA) according to the manufacturer's protocols. Viral RNA was isolated from the tissues using the Maxwell 16 LEV simplyRNA Tissue Kit (Promega) on a Maxwell 16 MDx instrument. A range of 20–40 mg of each tissue was homogenized using homogenization buffer from the Maxwell 16 LEV simplyRNA Tissue Kit, the TissueLyser (Qiagen, Hilden, Germany) and two 5 mm stainless steel beads (Qiagen, Hilden, Germany) in a 2 ml snap-cap tube, shaking twice for 3 min at 20 Hz each side. The isolation was continued according to the Maxwell 16 LEV simplyRNA Tissue Kit protocol, and samples were eluted into 50 μl RNase-free water. RNA was then quantified using qRT–PCR. If a tissue was negative by this method, a duplicate tissue sample was extracted using the Trizol Plus RNA Purification kit (Invitrogen). Because this purification kit allows for more than twice the weight of tissue starting material, there is an increased likelihood of detecting vRNA in tissues with low viral loads. RNA then was requantified using the same quantitative RT–PCR assay. Viral load data from plasma are expressed as vRNA copies/ml. Viral load data from tissues are expressed as vRNA copies/mg tissue.

**Quantitative reverse transcription PCR.** For ZIKV-PR, vRNA from plasma and tissues was quantified by qRT–PCR using primers with a slight modification to those described by Lanciotti et al. to accommodate African lineage ZIKV sequences[54]. The modified primer sequences are: forward 5′-CGYTGCCCAACA-CAAGG-3′, reverse 5′-CACYAAYGTTCTTTTGCABACAT-3′, and probe 5′-6fam-AGCCTACCTTGAYAAGCARTCAGACACYCAA-BHQ1-3′. The RT–PCR was performed using the SuperScript III Platinum One-Step Quantitative RT–PCR system (Invitrogen, Carlsbad, CA) on a LightCycler 480 instrument (Roche Diagnostics, Indianapolis, IN). The primers and probe were used at final concentrations of 600 nm and 100 nm respectively, along with 150 ng random primers (Promega, Madison, WI). Cycling conditions were as follows: 37 °C for 15 min, 50 °C for 30 min and 95 °C for 2 min, followed by 50 cycles of 95 °C for 15 s and 60 °C for 1 min. Viral RNA concentration was determined by interpolation onto an internal standard curve composed of seven 10-fold serial dilutions of a synthetic ZIKV RNA fragment based on a ZIKV strain derived from French Polynesia that shares > 99% similarity at the nucleotide level to the Puerto Rican strain used in the infections described in this manuscript.

**Macaque necropsy.** Following infection with ZIKV-PR via sc inoculation or mosquito-bite, macaques were sacrificed at ~15 days post-feeding or post needle inoculation for all animals. Tissues were carefully dissected using sterile instruments that were changed between each organ and tissue type to minimize possible cross contamination. Unfortunately, tissues from one sc-inoculated animal (634675) were not collected with sterile instruments and was therefore excluded from tissue analysis. For each of the two remaining animals infected by sc inoculation (566628 and 311413) and all four animals infected by mosquito bite, each organ/tissue was evaluated grossly in situ, removed with sterile instruments, placed in a sterile culture dish, weighed, and further processed to assess viral burden and tissue distribution or banked for future assays. Sampling of key organ systems suspected as potential tissue reservoirs for ZIKV and associated biological samples included the CNS (brain and eyes), urogenital, hematopoietic, and respiratory systems. A comprehensive listing of all specific tissues collected and analyzed is presented in Table 1.

**Deep sequencing.** Virus populations replicating in macaque plasma or mosquito saliva were sequenced in duplicate using a method adapted from Quick et al.[55]. Viral RNA was isolated from mosquito saliva or plasma using the Maxwell 16 Total Viral Nucleic Acid Purification kit, according to manufacturer's protocol. Viral RNA then was subjected to RT–PCR using the SuperScript IV Reverse Transcriptase enzyme (Invitrogen). Input viral RNA ranged from 294 to 64745 viral RNA templates per cDNA reaction (Supplementary Table 3). The cDNA was then split into two multiplex PCR reactions using the PCR primers described in Quick et. al with the Q5 High-Fidelity DNA Polymerase enzyme (New England Biolabs, Inc., Ipswich, MA). PCR products were tagged with the Illumina TruSeq Nano HT kit and sequenced with a 2 × 300 kit on an Illumina MiSeq.

**Sequence analysis.** Amplicon data were analyzed using a workflow we term "Zequencer 2017" (https://bitbucket.org/dhoconno/zequencer/src). Briefly, R1 and R2 fastq files from the paired-read Illumina miSeq dataset were merged, trimmed, and normalized using the bbtools package (http://jgi.doe.gov/data-and-tools/bbtools) and Seqtk (https://github.com/lh3/seqtk). Bbmerge.sh was used to merge reads, and to trim primer sequences by setting the forcetrimleft parameter 22. All other parameters are set to default values. These reads were then mapped to the reference amplicon sequences with BBmap.sh. Reads substantially shorter than the amplicon were filtered out by reformat.sh (the minlength parameter was set to the length of the amplicon minus 60). Seqtk was used to subsample to 1000 reads per amplicon. Quality trimming was performed on the fastq file of normalized reads by bbmap's reformat.sh (qtrim parameter set to 'lr', all other parameters set to default). Novoalign (http://www.novocraft.com/products/novoalign/) was used to map each read to ZIKV-PRVABC59 reference sequence KU501215. Novoalign's soft clipping feature was turned off by specifying the parameter "-o FullNW".

Approximate fragment length was set to 300 bp, with a s.d. of 50. We used Samtools to map, sort, and create an mpileup of our reads (http://samtools.sourceforge.net/). Samtools' base alignment quality (BAQ) computation was turned off; otherwise, default settings were used. SNP calling was performed with VarScan's mpileupcns function (http://varscan.sourceforge.net/). The minimum average quality was set to 30; otherwise, default settings were used. VCF files were annotated using SnpEff[56]. Accurate calling of end-of-read SNPs are a known weakness of current alignment algorithms[57]; in particular, Samtools' BAQ computation feature is known to be a source of error when using VarScan (http://varscan.sourceforge.net/germline-calling.html). Therefore, both Novoalign's soft clipping feature and Samtools' BAQ were turned off to increase the accuracy of SNP calling for SNPs occurring at the end of a read.

**Evolutionary analysis.** The π statistic, which estimates pairwise nucleotide diversity over a specified sequence length (in this case, individual ZIKV genes) without regard to a reference, was calculated using the Variance-at-position script in the open-source PoPoolation 1.2.2[58]. Synonymous and non-synonymous nucleotide diversity (πN and πS) were calculated using the PoPoolation script synnonsyn-at-position. For all calculations, minimum coverage was set to 100 and corrections were disabled.

**ZIKV NS1-specific ELISA.** IgM ZIKV-specific antibody responses were assessed using the Euroimmun diagnostic kit assay. Briefly, a 1:100 dilution of macaque serum was performed in duplicate and added to the precoated plates. The assay was performed following the manufacturer's instructions, with photometric measurements taken at 450 nm.

**Statistical analyses.** The difference in time to peak viremia between animals sc-inoculated and animals infected by mosquito bite was assessed with a linear mixed-effects model using the nlme package in R v3.3.2[59] with time to peak viremia as the response variable, route of infection as the explanatory variable, and animal ID as the random effect. The difference in peak viral loads between the two groups was analyzed with GraphPad Prism software and an unpaired Student's t-test was used to determine significant differences in viral loads. Differences in overall nucleotide diversity (π) among study groups was analyzed in GraphPad Prism using one-way analysis of variance with correction for multiple comparisons. Differences in synonymous and non-synonymous diversity (πN and πS) within groups were analyzed using Student's paired t-test in GraphPad Prism.

**Data availability.** Primary data that support the findings of this study are available at the Zika Open-Research Portal (https://zika.labkey.com). Zika virus/H.sapiens-tc/PUR/2015/PRVABC59-v3c2 sequence data have been deposited in the Sequence Read Archive (SRA) with accession code SRX2975259. The authors declare that all other data supporting the findings of this study are available within the article and its Supplementary Information files, or from the corresponding author upon request.

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

## Acknowledgements

The authors acknowledge Jens Kuhn and Jiro Wada for preparing silhouettes of macaques used in figures. We thank the Veterinary, Animal Care, Scientific Protocol Implementation, and the Pathology staff at the Wisconsin National Primate Research Center (WNPRC) for their contribution to this study. We also thank Emma Walker for assistance with the mouse experiments. Funding for this project came from DHHS/PHS/NIH R21AI131454 to M.T.A. and DHHS/PHS/NIH R01AI116382 to D.H.O. and from P51OD011106 awarded to the WNPRC, Madison, WI. This research was conducted in part at a facility that constructed with support from Research Facilities Improvement Program Grants RR15459-01 and RR020141-01. The publication's contents are solely the responsibility of the authors and do not necessarily represent the official views of the NCRR or NIH.

## Author contributions

M.T.A., T.C.F., D.M.D., D.H.O., J.E.O., S.L.O., and C.M.N. designed experiments. D.M. D., C.M.N., J.L., T.C.F., and M.T.A. analyzed data and drafted the manuscript. M.T.A. provided and prepared viral stocks, performed plaque assays, vector competence studies, mouse studies, and direct mosquito feeding studies. A.M.W., M.R.S., G.L.B., and T.C.F. developed and performed viral load assays. M.S.M., M.R.K., M.R.S., M.E.B., L.M.S., C.M.N., E.L.M., and D.M.D. coordinated and processed macaque samples for distribution. K.R.Z. and S.L.O. developed and performed the deep sequencing pipeline. M.R.K. maintained the Zika Open Portal site where data were stored and shared. N.S-D., E.P., S.C., and W.N. coordinated and performed macaque infections and sampling.

## Additional information

**Competing interests:** The authors declare no competing financial interests.

