## [Peer Review File · Nature Communications]

Reviewers' comments:

Reviewer #1 (Remarks to the Author):

This manuscript reports a proof-of-concept study that examined whether subcutaneous inoculation of ZIKV differs from mosquito bite delivery in a rhesus macaque model. In such studies the number of animals is typically very small (n=4 mosquito-infected animals and n=3 needle-infected animals) but the baseline information provided is valuable. The authors investigated the time course of viral load, tissue tropism, and within-host viral genetic diversity of a Puerto Rican strain of ZIKV.

1) One significant issue related to the nature of the study is that the virus inoculum injected by mosquito bite (from a varying number of mosquitoes) is not experimentally controlled. Therefore it is difficult to make a direct comparison with needle injection of a known inoculum (10^4 PFU). In other words the differences that the authors report between subcutaneous inoculation and mosquito bite delivery could be due to a difference in the inoculum size. The authors argue that 10^4 PFU is a dose likely to be delivered by a ZIKV-infected mosquito but this is based on mouse studies with WNV or in vitro transmission of DENV. When they measured the inoculum size of their ZIKV-infected mosquitoes by in vitro salivation, it ranged from $10^{1.5}$ to $10^{3.2}$ PFU per mosquito. A lower inoculum size by mosquito bite than needle injection could explain the delayed peak of viremia and higher inter-host variability in within-host viral genetic diversity. Although I understand that experimental studies of non-human primates are far from trivial, it would have been useful to include animals needle injected with different doses. This should at least be thoroughly discussed.

2) Another major issue is that interpretation of within-host viral genetic diversity patterns in mosquito saliva samples is very challenging. This is because the viral population initially from the viral stock went through a mouse infection followed by propagation in the mosquito that fed on the mouse. This is a multi-step journey with presumably many evolutionary forces involved. I strongly suggest that authors use more caution in their interpretation of deep sequencing data from single time points.

3) An important missing aspect is the time point of naïve mosquito challenge on ZIKV viremic monkeys. Nowhere in the manuscript could I find this critical information. The lack of infected mosquitoes may well be the result of the time point chosen. Please include this information and discuss it.

4) The authors state that "Mosquito transmission appeared to alter the frequency of SNPs" (lines 150-151) and further argue that one NS1 mutation may have been positively selected by mosquitoes. This is seemingly in contradiction with their conclusion that they "do not find evidence for strong natural selection" (line 187). Please clarify.

5) Description of the mosquito samples used for viral deep sequencing is confusing. The methods indicate that individual mosquito saliva samples were deep sequenced, however the authors also use the terms "infected mosquitoes" (line 148, line 169) and "mosquito

pool" (line 157, line 171) in the main text. Please clarify.

6) On lines 239-240, the authors state that subcutaneous inoculation "led to a similar tissue distribution as observed in mosquito-inoculated animals". This appears to be somewhat contradictory with their overall conclusion that tissue tropism depends on the infection route (lines 12-14).

7) Why was a random effect of the animal included in the analysis of time to peak viremia, but not in the analysis of peak viremia?

8) Following recent evidence on the role of secreted NS1 in ZIKV transmission to mosquitoes (Liu et al. Nature 2017), the authors may consider to measure the dynamics of plasmatic NS1 concentration in their macaque model.

Reviewer #2 (Remarks to the Author):

In their study "Infection via mosquito bite alters Zika virus replication kinetics in rhesus macaques", Dudley and colleagues introduce and define a novel NHP challenge model for Zika. The recent Zika outbreak has brought attention to the fact that adequate animal models to study this disease (and to test vaccines) were not readily available. Both, the commonly used IFNAR-KO mouse model and the pathogen challenge by injecting isolated virus with a needle are highly artificial. While the needle-based delivery of vector-borne pathogens (viruses, bacteria, parasites) continues to be widely used in the VBD-community, it introduces a lot of artifacts, many of which are mentioned in the manuscript. These caveats of needle-challenge approaches not only make it difficult to model diseases in animals and predict the usefulness of new vaccines, but also make it virtually impossible to compare results obtained in different labs (or even between different experiments). The authors described the much more relevant mosquito-bite challenge model, and, for the first time, in NHP. More importantly, they compared mosquito- and needle-delivered challenge side-by-side, and conducted a comprehensive analysis of both, host and virus. An obvious weakness of this, and many other NHP studies, is the number of animals in each group, which is further aggravated by the loss of tissue samples from one animal. Nevertheless, the results are still meaningful and helpful for the Zika community, but also for those studying the difference between vector- and needle-delivery of pathogens.

The following minor issues have been identified:

- In the section starting at line 134, the authors describe an extremely important evaluation of the effect of the transmission route on viral mutants. 1) It should be clarified what tissue the virus was obtained from for this analysis (this info is only in the Mat/Meth, but not mentioned in the results) and 2) it was surprising that this analysis (and the major findings) were completely omitted from the abstract. While the main purpose of the study was to establish the animal model, the findings related to the viral variants describe an important (additional) caveat and artifact of the needle-based challenge model and should, therefore, be mentioned in the abstract.

- The differences between the two experimental (NHP) groups in regards to the tissue distribution of virus are presented in a confusing and seemingly contradictory manner. This

can easily be addressed by re-wording the following statements/sections: in the Abstract (line 12), the authors suggest that the ZIKV tissue distribution is DIFFERENT between the two groups, but in line 117, the subtitle states that it is SIMILAR (same in line 239). Then, in line 246, the authors report evidence of ZIKV infection of brain and eye in needle-inoculated animals, while no brain/CNS infection could be detected after mosquito-bite, which indicates DIFFERENT tissue distribution! Please address these inconsistencies.

- Also confusing is the description of how reproductive tissues are affected: in line 125, the authors indicate that male reproductive tract tissues contained no detectable vRNA after mosquito bite. Starting the sentence that describes the absence of virus in these tissues in the SQ group (line 130) with "whereas" falsely implies that this is different in the mosquito-bite group. This entire paragraph would benefit from re-writing to clarify the differences between the two groups.
- Relying on colored symbols to differentiate between experimental groups/individual NHPs is not sufficient or helpful. The different groups should also be identifiable on b/w reproductions/printouts (without color, it is currently impossible to track vRNA copy numbers in individual animals in Fig. 1 and Fig.3, or tell the difference between the experimental groups in Fig. 4.)
- In Fig 4, the legend only describes the experimental groups, but does not explain what the different symbols (triangle, square, x and +) stand for. Since only the virus stock-group is shown in all three panels, it is not necessary to come up with a color scheme for the different experimental groups. Also, avoiding filled symbols would make it easier to see overlapping data points. The individual panels should be labeled (ie, each have a title) to indicate what experimental group is shown, and the order of the graphs should be changed (switch panel A and B) to allow for a better comparison of needle (sq) vs. mosquito-challenged animals.
- The authors were unable to detect viral RNA in male reproductive tract tissue, which is different from what has been reported in humans (albeit in a small number of individuals). This should be (briefly) addressed in the discussion.
- The discussion is quite comprehensive and addresses a variety of aspects of ZIKV infection. While the study's main objective obviously is the establishment of the animal model, the authors need to keep in mind that investigators would use it not only for studying Zika-pathology, but also for testing vaccines. A brief discussion of how the observed differences in the two models could affect the interpretation of protection-data in vaccination studies would be helpful. This is an issue that had previously triggered hot debates in the malaria field (the authors cite one of those studies) and it would be important to start this discussion for Zika.
- Minor formatting issue: All tables contain a large amount of unnecessary white space – reducing the height of (all) rows will also make the tables easier to read.

Reviewer #3 (Remarks to the Author):

The authors present data to support the hypothesis that infection of Rhesus macaques with Zika virus via the bite of an infected mosquito significantly alters the natural history of infection when compared to needle inoculation and they argue that the resultant model will

be valuable for studying ZIKV disease because it more closely resembles a natural exposure. The manuscript is well written and the data supports the conclusions that are discussed. Despite the low number of animals used per group the statistical methods are sufficient to support significant differences. The studies are novel for Zika virus and support conclusions drawn with other mosquito-borne viruses. I believe that this preliminary study will be valuable in directing future more robust studies to better understand the impact of mosquito transmission on Zika virus pathogenesis. However, the authors report that individual animals were fed on by as little as 5 or as much as 18 mosquitoes and they discuss the expected dose from a mosquito, but they don't discuss the impact of the cumulative dose from multiple feedings on the results. Similarly, on line 220 the authors state that different mosquitoes could transmit different founder populations but don't address the fact that each animal was fed on by multiple mosquitoes. Finally, the authors fail to consider the impact of amplification in immune deficient mice on the observed phenotypic diversity. Addressing these points would further strengthen the manuscript. Other things to consider: on line 27 replace "mouthparts" with "proboscis." On line 394 "Non or" should be replaced with "None of."

Thank you for the very thorough and timely review of this manuscript. Attached is the revised version of our article entitled “Infection via mosquito bite alters Zika virus replication kinetics in rhesus macaques” that has been modified in line with reviewers’ comments/concerns. The manuscript is greatly improved by incorporating these comments. Following is a point-by-point response to all concerns.

Reviewer 1:

1. One significant issue related to the nature of the study is that the virus inoculum injected by mosquito bite (from a varying number of mosquitoes) is not experimentally controlled. Therefore it is difficult to make a direct comparison with needle injection of a known inoculum (10^4 PFU). In other words the differences that the authors report between subcutaneous inoculation and mosquito bite delivery could be due to a difference in the inoculum size. The authors argue that 10^4 PFU is a dose likely to be delivered by a ZIKV-infected mosquito but this is based on mouse studies with WNV or in vitro transmission of DENV. When they measured the inoculum size of their ZIKV-infected mosquitoes by in vitro salivation, it ranged from $10^{1.5}$ to $10^{3.2}$ PFU per mosquito. A lower inoculum size by mosquito bite than needle injection could explain the delayed peak of viremia and higher inter-host variability in within-host viral genetic diversity. Although I understand that experimental studies of non-human primates are far from trivial, it would have been useful to include animals needle injected with different doses. This should at least be thoroughly discussed.

We agree and appreciate the acknowledgement that it is difficult to perform these types of experiments using nonhuman primates. We do intend to explore this in future studies and to hopefully quantify the exact dose of ZIKV delivered by a probing and feeding mosquito. To directly address this concern in the manuscript, we have added the following paragraph to the discussion:

“It also is possible that the differences in infection kinetics, tissue distribution, and viral population diversity in mosquito-bitten animals relevant to sc-inoculated animals may be due to the fact that feeding and probing mosquitoes deliver variable inoculum doses. We estimated the infectious ZIKV dose delivered by a mosquito to be $10^{1.5}$ to $10^{3.2}$ PFU per mosquito. In addition, each animal received multiple mosquito bites. Still, with our observed transmission frequency at 25% (**Table 1**) it is likely that each animal received a cumulative dose lower than 1×10^4 PFU, i.e., lower than the dose delivered by needle inoculation. However, as mentioned previously, collection of mosquito saliva via capillary feeding does not allow mosquitoes to probe and feed naturally and therefore likely underestimates the dose of virus inoculated. In fact, mosquitoes probing and feeding on a living host delivered doses of other flaviviruses that were 10- to 1000-fold higher than those measured using *in vitro* methods¹². Therefore, it also

is possible that the mosquito-bitten macaques received a cumulative dose that was several orders of magnitude higher than that delivered by sc-inoculation, but this requires further experimental verification. Importantly, the highest plasma viral loads were observed in the two animals (458001 and 268283) that received the least number of mosquito bites (five bites each). At the very least, these data warrant further exploration into the exact dose of ZIKV delivered by a feeding and probing mosquito.”

2. Another major issue is that interpretation of within-host viral genetic diversity patterns in mosquito saliva samples is very challenging. This is because the viral population initially from the viral stock went through a mouse infection followed by propagation in the mosquito that fed on the mouse. This is a multi-step journey with presumably many evolutionary forces involved. I strongly suggest that authors use more caution in their interpretation of deep sequencing data from single time points.

We agree with the reviewer and did not intend to give the impression that our study has conclusively characterized evolutionary processes shaping ZIKV populations in mosquitoes. Our analyses of diversity data show that, at the timepoints we tested, there is evidence for purifying selection on some viral genes, which is consistent with previous findings in arboviral infections, but this experiment was not designed to rigorously characterize the selective pressures encountered by virus populations throughout host transmission cycles. Indeed, in the discussion of our sequence data we state “our study investigated virus populations at single timepoints in infected macaques and mosquitoes and was therefore not powered to carefully evaluate within-host selection on virus populations over time” (line numbers: 237-239).

That said, we believe it is informative to use sequence data to determine whether, e.g., particular viral variants are consistently selected in mosquito-bitten vs. sc-inoculated animals; the data suggest that there are no such variants during early infection when viral population sizes are highest and selection would be expected to be most efficient.

The “reshuffling” of SNP frequencies we observe in mosquito-infected monkeys is indeed consistent with the multi-step journey under the influence of different selection pressures that the reviewer describes. To emphasize this point, and to underscore that we do not know where in the chain from mouse through mosquito to monkey such forces might be acting, we have replaced the phrase “mosquito transmission” in the results and discussion sections with phrases that more carefully note the multi-step process involved (line numbers: 152, 190-191, 221, and 232-233). We also added the phrase “in our data” to underscore that our interpretations are based on the available data and subject to the limitations described above.

3. An important missing aspect is the time point of naïve mosquito challenge on ZIKV viremic monkeys. Nowhere in the manuscript could I find this critical information. The lack of infected mosquitoes may well be the result of the time point chosen. Please include this information and discuss it.

We apologize for this omission and agree this is an important point. This information has been included (see line numbers: 198, 199, 436-437, and 761-763).

4. The authors state that “Mosquito transmission appeared to alter the frequency of SNPs” (lines 150-151) and further argue that one NS1 mutation may have been positively selected by mosquitoes. This is seemingly in contradiction with their conclusion that they “do not find evidence for strong natural selection” (line 187). Please clarify.

It is not our intention to argue that the mutation in NS1 encoding M220T described in the results section is positively selected and apologize if this was unclear. Rather, this mutation is provided as an example of a specific polymorphism that changes in frequency in viral populations as they pass from mice through mosquitoes to macaques. This SNP is notable because it is present at the consensus level in

multiple mosquito-bitten macaques, but as the reviewer notes, this fact alone is not sufficient to conclude that it is being positively selected. To clarify this, we have added the following sentence after the description of changing SNP frequencies: “We cannot determine from our data whether these changes in SNP frequencies are the result of natural selection or other processes, like genetic drift or founder effects.” (Line number: 160-162)

5. Description of the mosquito samples used for viral deep sequencing is confusing. The methods indicate that individual mosquito saliva samples were deep sequenced, however the authors also use the terms “infected mosquitoes” (line 148, line 169) and “mosquito pool” (line 157, line 171) in the main text. Please clarify.

We apologize. To clarify, the mosquito samples used for deep sequencing only consisted of saliva from individual mosquitoes that fed on mice infected with the same virus stock and were verified as transmission-competent in our *in vitro* assay. The terms “infected mosquitoes” and “mosquito pools” have been replaced with “mosquito saliva” or “mosquito saliva samples” where appropriate.

6. On lines 239-240, the authors state that subcutaneous inoculation “led to a similar tissue distribution as observed in mosquito-inoculated animals”. This appears to be somewhat contradictory with their overall conclusion that tissue tropism depends on the infection route (lines 12-14).

The inconsistencies have been addressed. The subtitle on line 117 has been changed from “similar” to “differs from”. The same change has been made in line 239. As the reviewer correctly suggests, these changes have been made to highlight the differences in tissue distribution between the two groups.

7. Why was a random effect of the animal included in the analysis of time to peak viremia, but not in the analysis of peak viremia?

Peak viral loads were analyzed using a student’s t-test, whereas time to peak viremia was analyzed using a mixed effects model. A t-test simply compares two groups’ mean and does not have random effects. The data for time to peak viral load were not appropriate for a simple t-test and therefore were analyzed using a mixed effects model that included the random effect of the animal identifier.

8. Following recent evidence on the role of secreted NS1 in ZIKV transmission to mosquitoes (Liu et al. Nature 2017), the authors may consider to measure the dynamics of plasmatic NS1 concentration in their macaque model.

We agree that further exploration into the dynamics of plasmatic NS1 concentration in macaques and the impact it might have on mosquito infection is relevant and interesting; however, it is beyond the scope of the work presented here and we hope to include this in future studies.

Reviewer 2:

1. In the section starting at line 134, the authors describe an extremely important evaluation of the effect of the transmission route on viral mutants. 1) It should be clarified what tissue the virus was obtained from for this analysis (this info is only in the Mat/Meth, but not mentioned in the results) and 2) it was surprising that this analysis (and the major findings) were completely omitted from the abstract. While the main purpose of the study was to establish the animal model, the findings related to the viral variants describe an important (additional) caveat and artifact of the needle-based challenge model and should, therefore, be mentioned in the abstract.

We have added to the results section that for the deep sequencing analysis virus was obtained from macaque plasma. This information was missing from the previous submission. Cell-culture produced

stock virus also was included in the analysis (new line numbers: 139). We have added this information to the abstract per the reviewer's suggestion.

2. The differences between the two experimental (NHP) groups in regards to the tissue distribution of virus are presented in a confusing and seemingly contradictory manner. This can easily be addressed by re-wording the following statements/sections: in the Abstract (line 12), the authors suggest that the ZIKV tissue distribution is DIFFERENT between the two groups, but in line 117, the subtitle states that it is SIMILAR (same in line 239). Then, in line 246, the authors report evidence of ZIKV infection of brain and eye in needle-inoculated animals, while no brain/CNS infection could be detected after mosquito-bite, which indicates DIFFERENT tissue distribution! Please address these inconsistencies.

The inconsistencies have been addressed. The subtitle on line 117 has been changed from "similar" to "differs from". The same change has been made in line 239. As the reviewer suggests, these changes have been made to highlight the differences in tissue distribution between the two groups.

3. Also confusing is the description of how reproductive tissues are affected: in line 125, the authors indicate that male reproductive tract tissues contained no detectable vRNA after mosquito bite. Starting the sentence that describes the absence of virus in these tissues in the SQ group (line 130) with "whereas" falsely implies that this is different in the mosquito-bite group. This entire paragraph would benefit from re-writing to clarify the differences between the two groups.

"Whereas" has been replaced with "likewise" to eliminate the false implication of a difference between the two groups. Aside from this change we respectfully disagree that the paragraph would benefit from re-writing. We believe the differences between the two groups are explicitly stated.

4. Relying on colored symbols to differentiate between experimental groups/individual NHPs is not sufficient or helpful. The different groups should also be identifiable on b/w reproductions/printouts (without color, it is currently impossible to track vRNA copy numbers in individual animals in Fig. 1 and Fig.3, or tell the difference between the experimental groups in Fig. 4.).

Figures 1 and 3 have been altered, so that now it is possible to track vRNA copy numbers in individual animals on black and white reproductions/printouts. Each animal has a unique symbol assigned to it, which is indicated on each figure. Likewise, figure 4 has been altered so symbols match those of Figures 1 and 3.

5. In Fig 4, the legend only describes the experimental groups, but does not explain what the different symbols (triangle, square, x and +) stand for. Since only the virus stock-group is shown in all three panels, it is not necessary to come up with a color scheme for the different experimental groups. Also, avoiding filled symbols would make it easier to see overlapping data points. The individual panels should be labeled (ie, each have a title) to indicate what experimental group is shown, and the order of the graphs should be changed (switch panel A and B) to allow for a better comparison of needle (sq) vs. mosquito-challenged animals.

We thank the reviewer for these comments and have made several changes to Figure 4 and the associated text. We now describe SNP distribution in mosquitoes first, then in mosquito-bitten monkeys, and finally in sc-inoculated monkeys. This allows us to place the panels showing SNP frequencies in mosquito- and sc-infected monkeys next to each other. We avoid filled symbols by using open or line symbols for all samples except the stock virus. Each panel now has a title and a legend showing which symbol corresponds to which sample. We retain a color scheme to preserve the colors associated with each group throughout the display items, but the revised Figure 4 will now render better in grayscale.

6. The authors were unable to detect viral RNA in male reproductive tract tissue, which is different from

what has been reported in humans (albeit in a small number of individuals). This should be (briefly) addressed in the discussion.

We are aware of no reports examining ZIKV tissue distribution in the male reproductive tract. There are several reports of Zika virus being shed in human semen for as long as six months; cases of male-to-female and male-to-male sexual transmission have also been reported. Both suggest that the male urogenital tract may serve as a reservoir for the virus. As a result, we have included the following statement in the discussion: “Finally, we did not detect ZIKV RNA in male reproductive tract tissues. This was somewhat surprising considering reports of both male-to-female and male-to-male sexual transmission of ZIKV ⁴¹, both of which are suggestive of the male urogenital tract tissues serving as potential reservoirs for the virus. It is possible that additional analyses of more animals at different timepoints using more careful sampling would more sensitively detect ZIKV RNA” (lines: 267-272).

7. The discussion is quite comprehensive and addresses a variety of aspects of ZIKV infection. While the study’s main objective obviously is the establishment of the animal model, the authors need to keep in mind that investigators would use it not only for studying Zika-pathology, but also for testing vaccines. A brief discussion of how the observed differences in the two models could affect the interpretation of protection-data in vaccination studies would be helpful. This is an issue that had previously triggered hot debates in the malaria field (the authors cite one of those studies) and it would be important to start this discussion for Zika.

We agree this is an important topic but are not aware of data comparing vaccine efficacy against flavivirus challenge delivered by mosquitoes vs. needles, so we are not comfortable speculating extensively on this topic. However, we agree that the biology of vector delivery may impact vaccine efficacy. To highlight that there is little information in this area for ZIKV, we have added the following statement to the discussion: “Whether mosquito-vectored ZIKV challenge might alter the efficacy of vaccines relative to needle inoculation remains unknown, but these results from other systems underscore the importance of studying pathogenic outcomes following natural exposure to a pathogen” (lines: 352-355)

8. Minor formatting issue: All tables contain a large amount of unnecessary white space – reducing the height of (all) rows will also make the tables easier to read.

The height has been reduced for all rows in each table.

Reviewer 3:

1. the authors report that individual animals were fed on by as little as 5 or as much as 18 mosquitoes and they discuss the expected dose from a mosquito, but they don’t discuss the impact of the cumulative dose from multiple feedings on the results.

See Reviewer 1, comment 1.

2. On line 220 the authors state that different mosquitoes could transmit different founder populations but don’t address the fact that each animal was fed on by multiple mosquitoes.

We added the following statement to the discussion: “Furthermore, because each macaque was bitten by multiple mosquitoes, viral populations in macaques could represent mixtures of populations from more than one mosquito” (Lines: 227-229)

3. The authors fail to consider the impact of amplification in immune deficient mice on the observed phenotypic diversity.

The reviewer correctly notes that ZIKV replicated in the absence of immune selection in IFNAR^{-/-} mice. It is therefore possible that large viral population sizes combined with a lack of immune selection allowed for viral diversification. It is further possible that some mutations could have emerged in mice that contributed to the phenotypic differences in virus replication kinetics and/or tissue distribution we observed in macaques infected by mosquito bite. Unfortunately we do not have access to samples from infected mice to characterize viral sequences in those animals. However, we believe it is unlikely that viral diversification in mice played a key role in the observed phenotypic differences in macaques: as mentioned in the text, no SNP or pattern of SNPs consistently distinguished viruses in mosquito-bitten monkeys from viruses infecting sc-inoculated animals. Therefore, to the extent that differences in replication kinetics or tissue distribution, which were consistently observed across mosquito-bitten monkeys, were genetically encoded in ZIKV populations, these differences were unlikely to be encoded by a single, common mutation or set of mutations.

It is formally possible that different viral genotypes could lead to similar phenotypes in different infected monkeys, and we agree that genetic differences in viral populations in mosquito-bitten monkeys could have functional impacts. However, given the lack of clear evidence for a consistently selected mutation or viral gene, and the well-documented impacts of natural vector transmission vs. needle inoculation on the outcomes of arboviral infection in other systems, we believe that mosquito bite per se, not viral genetic differences, is likely the most important factor driving the observed phenotypic differences in our study. To clarify this, we have added the following statement to the discussion: “Mosquitoes were infected by feeding on ZIKV-infected immune-deficient mice; it is therefore possible that viral variants emerging during replication in mice could have encoded some of the phenotypic differences we observed in macaques. However, this appears unlikely, as no specific SNP or constellation of SNPs was consistently detected that distinguished viruses replicating in mosquito-bitten animals from viruses present in sc-inoculated animals” (line numbers: 244-249).

4. On line 27 replace “mouthparts” with “proboscis.” On line 394 “Non or” should be replaced with “None of.”

These changes have been made.

REVIEWERS' COMMENTS:

Reviewer #1 (Remarks to the Author):

The authors have satisfactorily addressed the reviewers' concerns in the revised manuscript. In light of their responses they could however consider changing the main title to "Infection via mosquito bite alters Zika virus tissue tropism and replication kinetics in rhesus macaques".

Reviewer #2 (Remarks to the Author):

The authors have provided a thorough response to the reviewers' questions and concerns and have modified the manuscript accordingly. These modifications address the critiques appropriately and I have no further comments or suggestions.

Response to Reviewer's Comments

The authors have satisfactorily addressed the reviewers' concerns in the revised manuscript. In light of their responses they could however consider changing the main title to "Infection via mosquito bite alters Zika virus tissue tropism and replication kinetics in rhesus macaques".

We agree and have changed the title accordingly.